# Prediction Model of Hydropower Generation and Its Economic Benefits Based on EEMD-ADAM-GRU Fusion Model

**Jiechen Wang [1,\*], Zhimei Gao [2] and Yan Ma [1]**

1   School of Economics and Management, Northwest University, Xi'an 710127, China
2   School of System and Enterprises, Stevens Institute of Technology, Hoboken, NJ 07030, USA
\*   Correspondence: jiechen_wangl993@126.com

**Abstract:** As an important function of hydraulic engineering, power generation has made a great contribution to the growth of national economies worldwide. Therefore, it is of practical engineering significance to analyze and predict hydropower generation and its economic benefits. In order to predict the amount of hydropower generation in China and calculate the corresponding economic benefits with high precision, Ensemble Empirical Mode Decomposition (EEMD), Adaptive Moment Estimation (ADAM) and Gated Recent Unit (GRU) neural networks are integrated. Firstly, the monitoring data of hydropower generation is decomposed into several signals of different scales by the EEMD method to eliminate the non-stationary components of the data. Then, the ADAM optimization algorithm is used to optimize the parameters of the GRU neural network. The relatively stable component signals obtained from the decomposition are sent to the optimized GRU model for training and predicting. Finally, the hydropower generation prediction results are obtained by accumulating the prediction results of all components. This paper selects the time series of China's monthly power generation as the analysis object and forecasts the economic benefits by constructing the fusion prediction model. The RMSE EEMD-ADAM-GRU model is reduced by 16.16%, 20.55%, 12.10%, 17.97% and 7.95%, respectively, of compared with the NARNET, EEMD-LSTM, AR, ARIMA and VAR models. The results show that the proposed model is more effective for forecasting the time series of hydropower generation and that it can estimate the economic benefits quantitatively.

**Keywords:** hydraulic engineering; monitoring data; hydropower generation; forecast model; intelligent algorithm; deep learning

## 1. Introduction

At present, hydropower is an important renewable energy resource in the world [1–3]. The degree of hydropower development rate (developed hydropower generation/total hydropower generation that can be developed) in developed countries such as France and United States has reached 88% and 67%, respectively. China is rich in hydropower resources, but the degree of development is only 56%, which shows a certain gap with developed countries. Therefore, China's hydropower development will be in a growing trend in the future. Hydropower generation is at the core of hydropower development. The accurate prediction of hydropower generation can not only provide data support for the government's energy policy adjustment but also quantitatively evaluate the impact of power generation growth on the national economy [4,5].

Hydropower projects can be divided into daily regulation projects, monthly regulation projects and annual regulation projects according to their different regulation functions in the power system. Therefore, the variation trend of hydropower generation has significant periodic effects. At present, some researchers have carried out continuous research on the prediction of hydropower generation, using a variety of time series prediction models to simulate the change law of hydropower generation, such as regression analysis, gray

prediction, artificial neural network prediction, support vector machine, wavelet analysis prediction, etc. [6–9]. Although the above methods can predict the variation law of hydropower generation, the prediction accuracy for periodic signals is still not high [10–12].

With the continuous development of new artificial intelligence algorithms in recent years, a large number of studies have indicated that new artificial intelligence algorithms can be used to replace traditional prediction methods for periodic time series data [13–16]. Deep learning is a widely used direction in new artificial intelligence algorithms and has made remarkable achievements in search technology, data mining, machine learning and other fields [17–20]. The GRU neural network is an optimization model of the Long-Short Term Memory (LSTM) network and has proved its effectiveness and high accuracy in the field of water conservancy project prediction. GRU has proved that, compared with traditional machine learning algorithms, it has significant advantages in time series prediction, but the prediction accuracy of the single GRU neural network is greatly affected by the nonlinearity of the signal and the model parameters [21–23].

Aiming at solving the above problems in predicting the time series data with high-precision, combined prediction models are developed. Through the combination of different single forecasting models, the advantages of each model are complementary, to improve the prediction accuracy. Nath [24] combined the Ensemble Empirical Mode Decomposition (EEMD) with the LSTM model for groundwater radon precursor anomalies identification. The results indicated that the prediction accuracy of the EEMD-LSTM model was much higher than that of the LSTM model. EEMD is an improved empirical mode decomposition method, which can effectively mine different frequency components of signals. The combination with deep learning can improve the accuracy of deep learning prediction [25–29]. It can be indicated that the fusion method can be used for high-precision prediction of hydropower generation.

Therefore, aiming at the high-precision prediction of the time series of hydropower generation, this study proposes an intelligent fusion prediction model of hydropower generation based on EEMD-ADAM-GRU. The fusion model solves the problem that GRU model prediction is affected by signal nonlinearity and model parameters. Firstly, the EEMD decomposition method is used to reduce the nonlinearity of the original signal. Then, the ADAM optimization algorithm is used to optimize GRU model parameters. Finally, the optimized GRU model is used to simulate the time series variation of hydropower generation, and the validity and accuracy of the model are verified by comparing with multiple time series models. The data selects China's monthly hydropower generation from the years 2000 to 2022. Based on the national average electricity price, the power generation is converted into economic output value, and the impact of power generation on the national economic growth is calculated.

## 2. Methods

### 2.1. EEMD Model

The Ensemble Empirical Mode Decomposition (EEMD) is an improved method based on Empirical Mode Decomposition (EMD) [30–32]. The essence of the algorithm is to smooth the signal and decompose the change trend of different scales. EMD can accurately reflect the distribution law of the signal in time and frequency by deriving a series of intrinsic mode functions (IMF). At present, it is widely used for signal decomposition and denoising of nonlinear time series. It can effectively improve the prediction accuracy by combining it with the prediction model. The specific steps of EMD decomposition are as follows:

(1) Find out all local maximum values and minimum values contained in the original hydropower generation time series $y(t)$;

(2) Interpolate the local maximum values and minimum values to obtain the upper and lower envelopes $u(t)$ and $d(t)$;

(3) Calculate the average value $m(t)$ of the upper and lower envelope lines:

$$m(t) = \frac{u(t) + d(t)}{2} \tag{1}$$

(4) Get the new time series $h(t)$ that defines the difference between the original series $y(t)$ and the average series $m(t)$:

$$h(t) = y(t) - m(t) \tag{2}$$

(5) For different data sequences, it needs to be judged whether $h(t)$ is an intrinsic modulus function (IMF) or not. If the number of extreme points in $h(t)$ is equal to the number of zero crossing points, and the value of $m(t)$ is zero in any time, then $h(t)$ is an intrinsic modulus function. Otherwise, take $h(t)$ as the original sequence and repeat the above steps until the definition of intrinsic modulus function is satisfied.

(6) After calculating the first intrinsic modulus function $I_1(t)$, subtract $I_1(t)$ from the original sequence to obtain the residual value sequence $r_1(t)$:

$$r_1(t) = y(t) - I_1(t) \tag{3}$$

(7) The extraction process of the first intrinsic modulus function is completed. Then take $r_1(t)$ as a new original sequence and extract the intrinsic modulus function $I_n(t)$ in turn according to the above method. When $r_n(t)$ becomes a monotone sequence, its intrinsic modulus function cannot be extracted and the decomposition ends.

(8) The decomposition result of the original sequence is obtained by accumulating the decomposed components, which express as:

$$y(t) = \sum_{i=1}^{n} I_n(t) + r_n(t) \tag{4}$$

The EMD decomposition method performs well in non-stationary data processing, but there is the problem of mode mixing. It is found that adding noise to the original signal can solve the problem of mode mixing by taking advantage of the characteristics of the noise itself; thus, the EEMD method was proposed [33]. The improvement of the EEMD method is to put white Gaussian noise (selected for its randomness and uniform spectrum distribution) into the original signal, and then use the EMD method to decompose new signals. The EEMD method has good time-frequency resolution characteristics and can suppress mode mixing and solve mode splitting. In addition, the EEMD can accurately capture the local characteristics of the sequence and highlight the variation law of the subdivision frequency band. The decomposition process of the EEMD method is as follows.

(1) Add the white noise sequence $n(t)$ to the original hydropower generation time series $y(t)$, and get a new signal sequence $y'(t)$:

$$y'(t) = y(t) + n(t) \tag{5}$$

(2) The new signal sequence $y'(t)$ is decomposed by EMD to obtain the decomposition results as:

$$y'(t) = \sum_{i=1}^{n} I'_n(t) + r'_n(t) \tag{6}$$

(3) Add different white noises to the original signal, repeat the above steps for *N* times, obtain *N* groups of IMF components and residual components after EMD decomposition, and calculate the average value of N groups of IMF and residual components which are the EEMD decomposition results. The relevant expressions are shown in Equations (7) and (8).

$$IMF_i = \frac{1}{N} \sum_{j=1}^{N} I'_{ij}(t) \tag{7}$$

$$r(t) = \frac{1}{N}\sum_{j=1}^{N} r'_j(t) \tag{8}$$

where $I'_{ij}$ is the $i$-th IMF modal component obtained by the $j$-th decomposition, $j = 1, 2 \cdots N.r'_j$ is the residual component obtained by the $j$-th decomposition.

### 2.2. GRU Model

GRU is a kind of Recurrent Neural Network (RNN) [34,35]. RNN is mainly used to process and predict time series data, and it is effective for solving the relationship between time series values at different times. Figure 1 shows a typical RNN network unit. The RNN network consists of three layers: input layer, hidden layer and output layer. There are two ways to describe the unit: folding structure (left in Figure 1) and expanding structure (right in Figure 1).

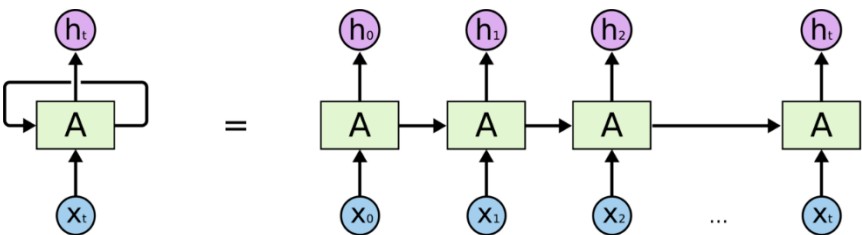

**Figure 1.** The typical RNN network unit.

At each time, RNN will give a current output information $h_t$ and update the network state based on the current input information $x_t$ and the current model state $A$. The input of RNN is not only from the input $x_t$, but also from a loop unit to provide the output of the hidden layer at the previous time.

The RNN neural network model is a kind of supervision model. The model needs to be trained with observed data. In the training process, the parameters of each neuron in the model are solved through the back propagation algorithm. The optimal model obtained by training retains the characteristic information of the data in the connection weight value, and the subsequent input of unlabeled data can derive the prediction results.

In recent years, with the continuous development of RNN neural networks, the structure of RNN neural networks has also undergone many changes, among which the long-short term memory (LSTM) network model has been widely used. LSTM adds three "gate structures" on the basis of RNN. It relies on these "gate structures" to selectively add past information to the current moment.

LSTM solves the problem of RNN gradient disappearance and has been widely used in the field of time series prediction. Some other variants have also been produced on the basis of LSTM "gate structure". GRU is one of the most recognized variants of LSTM. Compared with LSTM, GRU has fewer training parameters, while it outperforms LSTM on all tasks except for language modelling [36]. GRU is more effective and has only two gate units (update gate and reset gate).The internal unit structure of a GRU neural network is shown in Figure 2.

The calculation formula of GRU neural network is as follows:

$$\begin{cases} z_t = \sigma(W^{(z)}x_t + U^{(z)}h_{t-1}) \\ r_t = \sigma(W^{(r)}x_t + U^{(r)}h_{t-1}) \\ u_t = \tanh(r_t U h_{t-1} + W x_t) \\ h_t = (1 - z_t)u_t + z_t h_{t-1} \end{cases} \tag{9}$$

where $z_t, r_t$ are update gate and reset gate, $x_t$ is input information, $h_t$ is the output information, $u_t$ is the summary of the input $x_t$ and the previous hidden layer $h_{t-1}$, $W^{(z)}, U^{(z)}, W^{(r)}$, $U^{(r)}, U, W$ are the training parameter matrix, $\sigma$ and tanh are sigmoid functions and hyperbolic tangent functions.

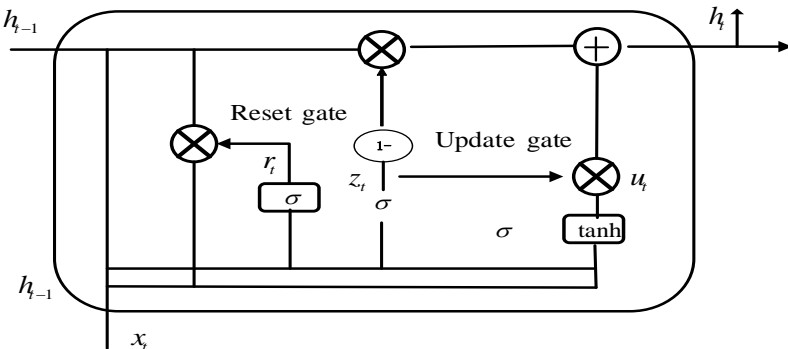

**Figure 2.** GRU internal unit structure diagram.

### 2.3. ADAM Optimization Algorithm

The conventional GRU neural network uses the random gradient descent algorithm to update the parameters of the neural network. The convergence speed of this model algorithm is slow in the early stage, and the problem of precision decline easily occurs. In order to improve the prediction accuracy and accelerate the convergence speed of the model, ADAM optimization algorithms are used to optimize the GRU neural network model [37,38].

The GRU model generally selects mean square error (MSE) as the loss function which can well express the error between the predicted value and the actual output value. ADAM designs independent adaptive learning rates for different parameters by calculating the first-order and second-order moment estimations of the gradient of loss function which are shown as follows:

$$m_t = \beta_1 \times m_{t-1} + (1 - \beta_1) \times dx \tag{10}$$

$$v_t = \beta_2 \times v_{t-1} + (1 - \beta_2) \times (dx)^2 \tag{11}$$

where $m_t, v_t$ are the first-order and second-order moment estimation at time $t$, $\beta_1, \beta_2$ are exponential decay rate of first-order moment estimation and second-order moment estimation (generally $\beta_1 = 0.9, \beta_2 = 0.999$), and $dx$ is the gradient of loss function.

The second step is to correct the loss function, which corrects the first and second moment estimates by the following equation:

$$m'_t = \frac{m_t}{1 - \beta_1} \tag{12}$$

$$v'_t = \frac{v_t}{1 - \beta_2} \tag{13}$$

GRU neural network parameters are updated as follows:

$$x_{t+1} = x_t - \frac{\alpha \times m'_t}{\sqrt{v'_t} + \varepsilon} \tag{14}$$

where $x_{t+1}, x_t$ are the parameter vectors at times $t + 1$ and $t$, $\varepsilon$ is a positive number close to zero, which can prevent the denominator from being zero, and $\alpha$ is the learning rate.

### 2.4. Construction of Fusion Prediction Model

Hydropower generation sequence usually presents an unstable state and is affected by the impact of power system regulation. In order to overcome the problem of the low prediction accuracy of a single model, this paper uses a fusion model of three methods to predict hydropower generation. The fusion model is composed of the EEMD, ADAM and GRU methods. Firstly, the EEMD method is used to split the original signal into signals of different scales, thus greatly reducing the non-stationary property of the hydropower signal. Secondly, the ADAM optimization algorithm is used to optimize the model parame-

ters of the GRU neural network, and the decomposed component signals are sent to the optimized GRU model for training to obtain their respective prediction results. Finally, the hydropower prediction results are obtained by accumulating all component results.

For the prediction of hydropower generation in China, the EEMD-ADAM-GRU time series model is introduced to predict its change law. The EEMD-ADAM-GRU model can simulate the input-output relationship between power generation and time and quantitatively analyze the time-varying law of hydropower generation. The construction of the prediction model includes four steps: data input, data normalization, data decomposition, model parameter selection, model prediction, and model accuracy evaluation. The flow chart of the prediction model is shown in Figure 3. The specific modeling steps are as follows:

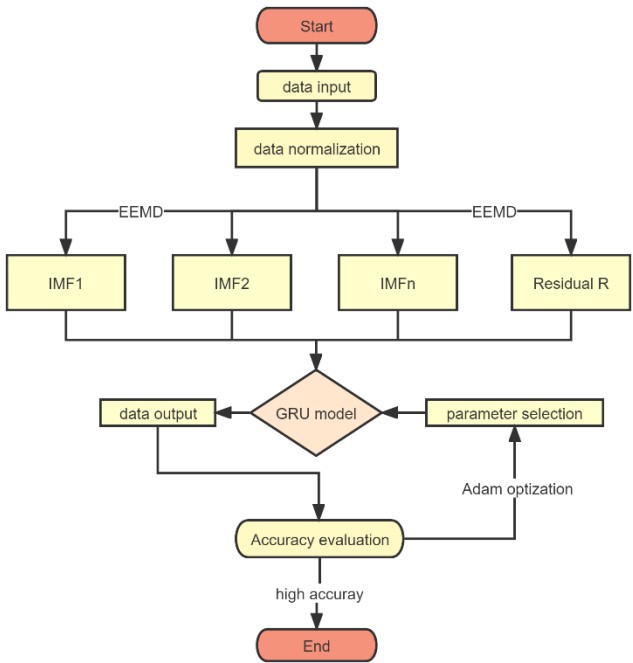

**Figure 3.** EEMD-ADAM-GRU construction flow chart.

(1) Data input: Obtain the historical data of China's hydropower generation, and divide the data into training set and test set.

(2) Data normalization: Normalization can improve the convergence speed and accuracy of the model. Linear function normalization is used to scale the data into the range of [0, 1]. The normalization formula is as follows:

$$x_{norm} = \frac{x - x_{\min}}{x_{\max} - x_{\min}} \tag{15}$$

where $x$ is the original data, $x_{\max}$ and $x_{\min}$ are the maximum and minimum values of the data, respectively.

(3) Data decomposition: As EEMD is an adaptive decomposition method, it does not need to set decomposition parameters in advance, so the EEMD method is used to split the results of the above steps to obtain IMF components of different scales and the residual component R.

(4) Model parameter selection: The prediction accuracy of the GRU model is related to three important parameters: the number of hidden layer neurons, the maximum number of iterations, and the learning rate. Therefore, it is necessary to constantly adjust the parameters through the ADAM optimization algorithm to obtain the model parameters corresponding to the highest prediction accuracy.

(5) Model prediction output: The data obtained in step3 are sent to GRU network for training, and all the predicted results are summed up to get the final prediction result of hydropower generation.

(6) Model precision evaluation: The prediction model evaluation index is used to measure the difference between the original data and the model prediction data. Two common indexes are introduced: Root Mean Square Error (RMSE) and Standard Deviation (SD). The specific formula is as follows:

$$RMSE = \sqrt{\frac{\sum\limits_{i=1}^{N}(x_i - \hat{x}_i)^2}{N}} \tag{16}$$

$$SD = \sqrt{\frac{\sum\limits_{i=1}^{N}(x_i - \overline{x})^2}{N-1}} \tag{17}$$

## 3. Case Study

The number of water conservancy projects and the total amount of water resources that can be developed in China are ranked first in the world. For the content of this study, China's research data is relatively easy to obtain. Therefore, China's water resources for power generation are selected as the research sample to quantitatively analyze the effectiveness and prediction accuracy of the proposed method. According to the data released by China's National Bureau of Statistics, China's power generation in total reached 8112.18 billion kWh in 2021. Of that total, the thermal power generation capacity is 5770.27 billion kWh, accounting for 71.13% of the total power generation capacity in China. The hydropower generation capacity is 1184.02 billion kWh, which is about 14.6% of the total national power generation capacity. With the rapid development of hydropower construction and hydropower technology, China's hydropower industry has made considerable progress. The hydropower generation capacity is growing gradually, and its proportion in the total power generation capacity is increasing.

This study selects the national hydropower generation amount from February 2000 to July 2022 (the data is from the National Bureau of Statistics, http://data.stats.gov.cn, 15 September 2022). The hydropower generation data is shown in Figure 4. The data is divided into training set and test set. The training set selects the data from October 2000 to May 2020, and the test set selects the data from June 2020 to July 2022. The training set and test set data are normalized by Equation (2).

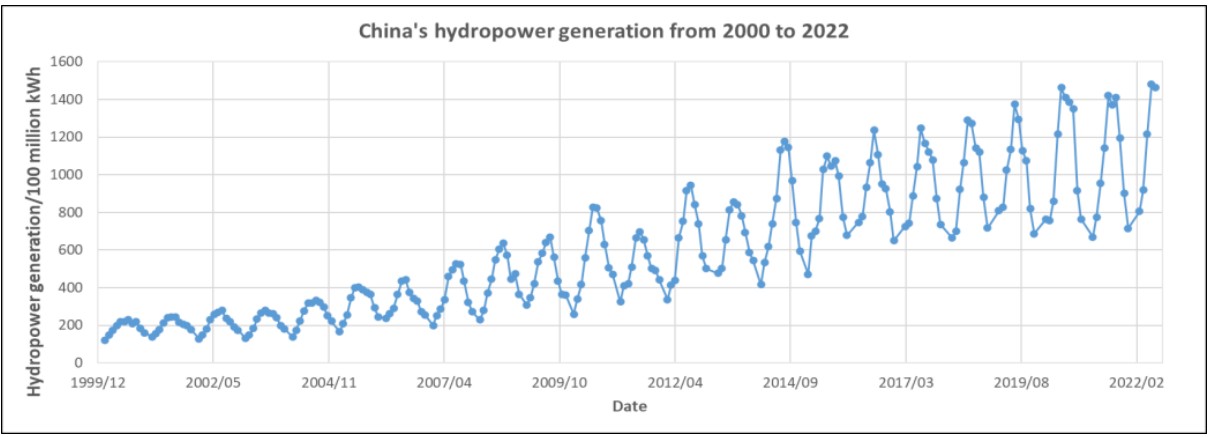

**Figure 4.** China's hydropower generation data from 2000 to 2022.

The original hydropower generation data series has high non-stationary and nonlinear characteristics, so it is difficult to directly predict the data series. Therefore, the EEMD method is used to process the original data to fully extract the characteristics of the original data. Figure 5 shows four IMF components and one residual component obtained from the original data after EEMD decomposition. The data decomposition can convert the original data into high- and low-frequency components with good stability and strong regularity. By replacing the original signal with components as the prediction input of GRU neural network model, the model can fully learn the information in the data during the training process.

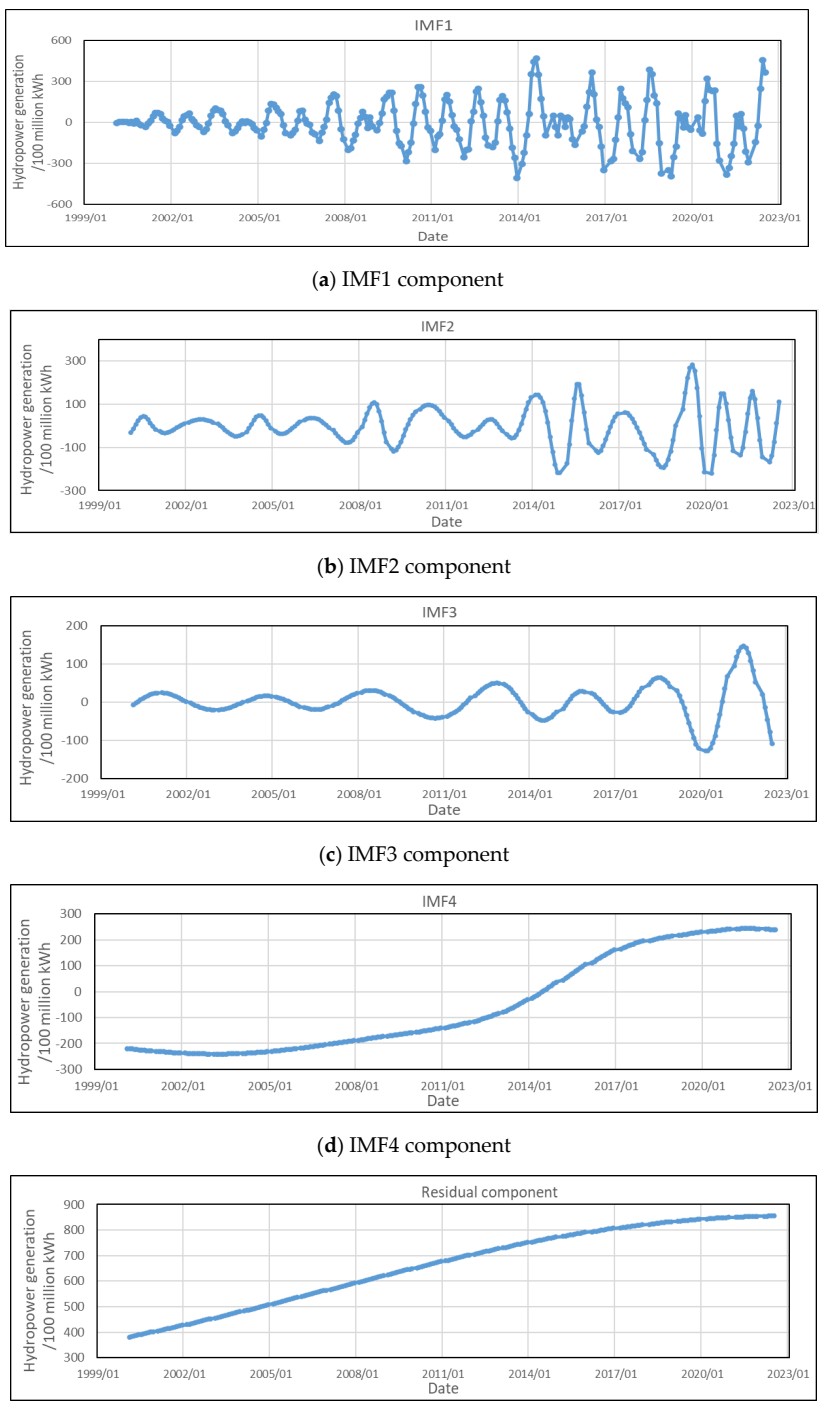

(**a**) IMF1 component

(**b**) IMF2 component

(**c**) IMF3 component

(**d**) IMF4 component

(**e**) Residual component

**Figure 5.** Decomposition results of China's hydropower generation data.

The selected data sequence is from February 2000 to July 2022. There are 248 groups of data. The first 80% of the data are selected as the training set, the last 10% as the validation set, to verify the optimal parameters of the model and estimate the training effect of the model, and the last 10% as the test set, to evaluate the prediction accuracy and generalization of the model. Based on the decomposition results of the training set, the different components of hydropower generation are dynamically simulated by the GRU neural network. In order to obtain the optimal model parameters and ensure the prediction accuracy, the ADAM optimizer is selected to optimize the GRU model parameters. The optimized GRU model parameters are shown in Table 1. Model parameters are important factors affecting the prediction accuracy of each time series model. Using the ADAM optimization algorithm, the model parameter values of each comparison model are obtained and shown in Table 2.

**Table 1.** GRU model parameters.

| Parameter | Value | Parameter | Value |
|---|---|---|---|
| Max epochs | 250 | Learn rate schedule | 'piecewise' |
| Gradient threshold | 1 | Learn rate drop period | 125 |
| Initial learn rate | 0.0005 | Learn rate drop factor | 0.2 |
| Number of hidden layers | 1 | Number of neurons | 128 |
| Active function | Sigmoid | Dropout | 0.2 |

**Table 2.** The parameter value of comparison models.

| Comparison Model | Parameter Title | Value |
|---|---|---|
| NARNET | Number of neurons | 30 |
| | Feedback mode | 'Open' |
| | Train function | 'trainlm' |
| LSTM | Max epochs | 250 |
| | Gradient threshold | 1 |
| | Initial learn rate | 0.0005 |
| | Learn rate schedule | 'piecewise' |
| | Learn rate drop period | 125 |
| | Learn rate drop factor | 0.2 |
| AR | Compound AR polynomial degree | 15 |
| | Compound MA polynomial degree | 0 |
| | Degree of nonseasonal integration | 0 |
| | Degree of seasonal differencing polynomial | 0 |
| ARIMA | Compound AR polynomial degree | 4 |
| | Compound MA polynomial degree | 8 |
| | Degree of nonseasonal integration | 1 |
| | Degree of seasonal differencing polynomial | 0 |
| VAR | Multivariate autoregressive polynomial order | 10 |

In order to verify the effectiveness of the EEMD-ADAM-GRU prediction model, the commonly used time series prediction models Nonlinear Autoregressive Neural Network (NARNET), Autoregression (AR), Autoregressive Integrated Moving Average (ARIMA), Vector Autoregression (VAR) are selected as comparison models. Also, EEMD-LSTM model is selected as comparison model to verify the prediction accuracy of EEMD combined with different depth learning algorithms. The Taylor plot of different models are shown in Figure 6, and the prediction evaluation indexes values of test set are shown in Table 3.

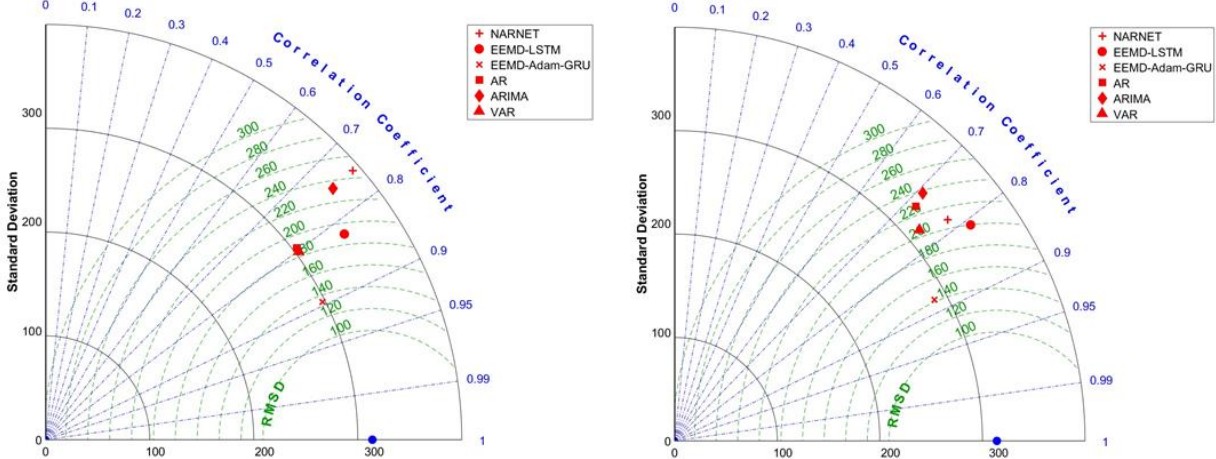

(**a**) The Taylor plot of the validation set      (**b**) The Taylor plot of the test set

**Figure 6.** Comparison of predicted values of different models.

**Table 3.** Comparison of prediction performance in test set.

| Model | RSME/Billion kWh | Decline Ratio | SD/billion kWh | Decline Ratio |
|---|---|---|---|---|
| NARNET | 179.34 | 16.16% | 324.32 | 15.69% |
| EEMD-LSTM | 189.24 | 20.55% | 338.32 | 19.18% |
| EEMD-ADAM-GRU | 150.35 | / | 273.42 | / |
| AR | 171.04 | 12.10% | 310.43 | 11.92% |
| ARIMA | 183.28 | 17.97% | 323.56 | 15.50% |
| VAR | 163.33 | 7.95% | 298.21 | 8.31% |

It can be seen from Figure 6 and Table 3 that the RMSE and SD of EEMD-ADAM-GRU model are the smallest; the RMSE of EEMD-ADAM-GRU model is reduced by 16.16%, 20.55%, 12.10%, 17.97% and 7.95%, respectively, compared with the NARNET, EEMD-LSTM, AR, ARIMA and VAR models; and the SD of EEMD-ADAM-GRU model is reduced by 15.69%, 19.18%, 11.92%, 15.50% and 7.95%, which shows that the EEMD-ADAM-GRU model has the highest prediction accuracy for the time series of hydropower generation. The GRU model is the representative of RNN and can predict the time series with high accuracy, especially for the time series with periodic non-stationary fluctuations such as hydropower generation. Therefore, the proposed prediction model can provide reliable data support for the analysis of the hydropower industry.

In order to study the change law of economic benefits brought by hydropower generation, the definition of economic benefits of hydropower generation needs to be clarified. The economic benefits brought by hydropower generation mainly include two kinds: ① Direct economic benefits: the economic benefits obtained from the sale of power generation are the most basic benefits of hydropower and the key to evaluating the project benefits. ② Indirect economic benefits: economic benefits from irrigation, water supply, shipping, etc. As the direct economic benefit is the key to evaluating the project benefit and is easy to calculate directly, this paper selects it as the object for evaluating the economic benefit of hydropower generation. The total benefits can be obtained by multiplying the generating capacity by the unit price. Since the hydropower unit price is differentiated in different provinces, and there is no direct statistical data on the sales price of different hydropower stations, it takes 0.36 yuan/kWh, which is the average unit price in China as hydropower unit price. Therefore, based on the EEMD-ADAM-GRU prediction model, the direct economic benefits of hydropower generation in the year from August 2022 to August 2023 are predicted. The

results are shown in Figure 7, which lists the output results of economic benefits from June 2020 to August 2023.

**Figure 7.** Predicted values of economic benefits of hydropower generation in China.

It can be seen from Figure 7 that the economic benefit curve of hydropower generation has obvious periodicity, which is mainly due to the seasonal effect of hydropower generation. In summer, there is more rainfall and large hydropower generation capacity, resulting in significant economic benefits. In the dry season in winter, the hydropower generation is small and the economic benefit is low. With the continuous increase of the total installed capacity of hydropower stations in China, the overall hydropower generation capacity shows a trend of periodic and gradual increase. The minimum and maximum value of annual power generation benefits are increasing year by year, and the monthly economic benefits of hydropower generation fluctuate between 20 billion yuan and 60 billion yuan. The economic benefits of hydropower generation in China are predicted to reach 40.458 billion yuan in August 2023, and the cumulative economic benefits for the year from August 2022 to August 2023 will reach 482.525 billion yuan. It indicates that hydropower generation has brought significant economic benefits in China.

## 4. Conclusions

As an important function of water conservancy projects, quantitative analysis of the future trend of hydropower generation and evaluation of its economic benefits have important engineering significance. Aiming at this practical problem, this paper proposes a high-precision prediction model of hydropower generation based on an EEMD-ADAM-GRU neural network. This fusion model makes full use of the GRU neural network's advantages in predicting periodic time series and excavates in depth the different components of hydropower generation by the EEMD decomposition method to improve the prediction accuracy of the GRU model. Through case analysis, the prediction performance of the proposed model is compared with that of several common time series models, and the following conclusions are derived:

(1) As an improved RNN neural network, the GRU model can conduct deep mining on the change rule of time series and derive high-precision prediction accuracy on the time series of hydropower generation. EEMD signal decomposition and ADAM model parameter optimization can further improve the prediction accuracy of the GRU model. Compared with the conventional time series prediction model, the average prediction error is reduced by more than 10%.

(2) The economic benefits of hydropower generation measure only the direct economic benefits. In addition, the hydropower price is replaced by the national average hydropower unit price, which does not reflect the fluctuation of hydropower prices with months. It should study the pricing mechanism and fluctuation law of hydropower prices in all provinces of the country in the future, so as to accurately measure the economic benefits of hydropower generation.

(3) The proposed prediction model is based on the concept of the "signal decomposition +parameter optimization+ prediction model". There are many types of signal decomposition methods and prediction models in existing research results. This paper only selects representative methods to combine and analyze. In future research, it is necessary to expand the model selection and analyze the prediction accuracy of different model combinations to propose a more universal prediction method.

**Author Contributions:** Conceptualization, J.W. and Z.G.; methodology, J.W. and Y.M.; software, J.W.; validation, J.W. and Z.G.; formal analysis, J.W.; investigation, J.W.; resources, J.W. and Y.M; data curation, J.W.; writing—original draft preparation, J.W.; writing—review and editing, J.W.; visualization, J.W.; supervision, J.W.; project administration, J.W. All authors have read and agreed to the published version of the manuscript.

**Funding:** This research received no external funding.

**Institutional Review Board Statement:** Not applicable.

**Informed Consent Statement:** Not applicable.

**Data Availability Statement:** Data openly available in a public repository.

**Conflicts of Interest:** The authors declare no conflict of interest.

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
