# Peer review of "Prediction Model of Hydropower Generation and Its Economic Benefits Based on EEMD-ADAM-GRU Fusion Model"

_water, doi:10.3390/w14233896_

Round 1
Reviewer 1 Report
This subject addressed is within the scope of the journal. However, the manuscript in the present version contains several problems. Appropriate revisions should be undertaken in order to justify recommendation for publication.
1. EEMD decomposition method is used for decomposition to capture data noise. why? How will this affect the results? More details should be furnished.
2. It is mentioned that GRU is used as main model. What are the advantages of adopting this particular method over others in this case? How will this affect the results? More details should be furnished. Why not tried advanced hybrid models for comparison? For example,LSTM-ALO,DENFIS,GMDH,OP-ELM,LSSVM-GSA recently used in literature. Should add these models recent literature and also explain why not adopted those advanced version?
3. For readers to quickly catch your contribution, it would be better to highlight major difficulties and challenges, and your original achievements to overcome them, in a clearer way in abstract and introduction.
4. It is mentioned that CHINA is adopted as the case study. What are other feasible alternatives? What are the advantages of adopting this case study over others in this case? How will this affect the results? The authors should provide more details on this.
5. There is a serious concern regarding the novelty of this work. What new has been proposed?
6. Abstract needs to modify and to be revised to be quantitative. You can absorb readers' consideration by having some numerical results in this section.
7. There are some occasional grammatical problems within the text. It may need the attention of someone fluent in English language to enhance the readability.
8. Since the some figures have low-resolution printing, the reviewer cannot recognize them clearly. Please revise them with high resolution.
9. The discussion section in the present form is relatively weak and should be strengthened with more details and justifications.
10. In conclusion section, limitations and recommendations of this research should be highlighted.
11. The authors have to add the state-of-the art references in the manuscripts.
https://doi.org/10.3390/en14123643
https://doi.org/10.1016/j.seta.2022.101962
https://doi.org/10.1016/j.egyr.2022.09.015
https://doi.org/10.3390/math10162971
https://doi.org/10.1007/s00477-018-1560-y
https://doi.org/10.1016/j.jhydrol.2022.127427
12. Some key parameters are not mentioned. The rationale on the choice of the set of parameters should be explained with more details. Have the authors experimented with other sets of values? What are the sensitivities of these parameters on the results?
13. It is mentioned that two performance indexes were used. What are the advantages of adopting these indexes over others (MAE, NSE, willimot index) in this case? How will this affect the results? More details should be furnished.
14. Why not draw Taylor and violin plots to compare the results?
Author Response
Dear editors and reviewers:
Thank you for your letter and the reviewers’comments on our manuscript entitled " Prediction model of economic benefits from hydropower generation based on EEMD-Adam-GRU". Those comments are very helpful for revising and improving our paper, as well as the important guiding significance to other research. We have studied the comments carefully and made corrections which we hope meet with approval. The main corrections are in the manuscript and the responds to the reviewers’comments are as follows.
Replies to the reviewers’comments:
Reviewer #1:
- EEMD decomposition method is used for decomposition to capture data noise. why? How will this affect the results? More details should be furnished.
Response:EEMD is an improved empirical mode decomposition method, which can effectively mine different frequency components of signals. And the combination with deep learning can improve the accuracy of deep learning prediction. In the introduction, the above statement and literature support are added.
- It is mentioned that GRU is used as main model. What are the advantages of adopting this particular method over others in this case? How will this affect the results? More details should be furnished. Why not tried advanced hybrid models for comparison? For example,LSTM-ALO,DENFIS,GMDH,OP-ELM,LSSVM-GSA recently used in literature. Should add these models recent literature and also explain why not adopted those advanced version?
Response:Deep learning is a new research direction in the field of machine learning in recent years. With its complex data mining structure, deep learning has been widely used in the field of data prediction. GRU is a deep neural network improved and optimized on the basis of LSTM. It has faster convergence speed and maintains the accuracy close to LSTM. GRU has proved that compared with traditional machine learning algorithms, It has significant advantages in time series prediction.
- For readers to quickly catch your contribution, it would be better to highlight major difficulties and challenges, and your original achievements to overcome them, in a clearer way in abstract and introduction.
Response:I have revised the abstract and introduction part so as to highlight major difficulties, challenges and my original achievements in a clearer way.
- It is mentioned that CHINA is adopted as the case study. What are other feasible alternatives? What are the advantages of adopting this case study over others in this case? How will this affect the results? The authors should provide more details on this.
Response:The number of water conservancy projects and the total amount of water resources that can be developed in China are ranked first in the world. For the content of this study, China's research data is relatively easy to obtain. Therefore, China's water resources power generation is selected as the research sample to quantitatively analyze the effectiveness and prediction accuracy of the methods proposed in this paper.
- There is a serious concern regarding the novelty of this work. What new has been proposed?
Response:There are three main innovative work contents in this paper. (1) This paper selects hydropower generation and economic benefits as the research object. Since hydropower is one of the most important functions of water conservancy projects, and the estimation of its economic benefits of power generation is an important way to evaluate the benefits of water conservancy projects, there are few literatures on this issue in the past. (2) This paper proposes a high-precision prediction model for time series, which is the first attempt to integrate EEMD, Adam and GRU models. (3) The calculation results of the case analysis can roughly estimate the future power generation benefits of water conservancy projects, and can provide a reference basis for operation managers and investors.
- Abstract needs to modify and to be revised to be quantitative. You can absorb readers' consideration by having some numerical results in this section.
Response:The quantitative analysis of the prediction results of the EEMD-Adam-GRU model and the comparison model is added in the summary to better express and verify the effectiveness of the proposed model.
- There are some occasional grammatical problems within the text. It may need the attention of someone fluent in English language to enhance the readability.
Response: I have corrected the grammatical errors and some expression errors in the text.
- Since some figures have low-resolution printing, the reviewer cannot recognize them clearly. Please revise them with high resolution.
Response: I have modified and replaced some unclear pictures.
- The discussion section in the present form is relatively weak and should be strengthened with more details and justifications.
Response: I have carefully revised the part of the conclusion that needs to be strengthened.
- In conclusion section, limitations and recommendations of this research should be highlighted.
Response: I add the statement in conclusion section that express the limitations and recommendations of this research that “The proposed prediction model is based on the concept of "signal decomposition +parameter optimization+ prediction model". There are many types of signal decom-position methods and prediction models in existing research results. This paper only selects representative methods to combine and analyze. In future research, it is necessary to expand the model selection, and analyze the prediction accuracy of different model combinations to propose a more universal prediction method.”
- The authors have to add the state-of-the art references in the manuscripts.
Response: I have added the state-of-the art references in the manuscripts.
- Some key parameters are not mentioned. The rationale on the choice of the set of parameters should be explained with more details. Have the authors experimented with other sets of values? What are the sensitivities of these parameters on the results?
Response: Model parameters are important factors affecting the prediction accuracy of each time series model. Using Adam optimization algorithm, the model parameter values of each comparison model are obtained as shown in Tab.2. By using the optimization algorithm, the influence caused by random selection of model parameters can be avoided.
- It is mentioned that two performance indexes were used. What are the advantages of adopting these indexes over others (MAE, NSE, willimot index) in this case? How will this affect the results? More details should be furnished.
Response: The Root Mean Square Error (RMSE) and Standard Deviation (SD) are more often used as evaluation indicators in the performance analysis of prediction models. Therefore, this paper replaces the evaluation indicators with RMSE and SD which is consistence with Taylor plot index.
- Why not draw Taylor and violin plots to compare the results?
Response: Taylor and violet plots is an effective method to intuitively display the model error. The original diagram is intended to show the change trend of prediction performance of different models. Therefore, the graph is replaced by the Taylor plot.
Reviewer 2 Report
Dear Authors,
Please see the referee report attached.
Best regards

Author Response
Dear editors and reviewers:
Thank you for your letter and the reviewers’comments on our manuscript entitled " Prediction model of economic benefits from hydropower generation based on EEMD-Adam-GRU". Those comments are very helpful for revising and improving our paper, as well as the important guiding significance to other research. We have studied the comments carefully and made corrections which we hope meet with approval. The main corrections are in the manuscript and the responds to the reviewers’comments are as follows.
Replies to the reviewers’comments:
Reviewer #2:
Major
- Some concerns can be raised about a potential overfitting of the test set via hyperparameter tuning. Current best practice is to define a validation set in addition to the training and test set and perform Stacked Cross-Validation to optimize hyperparameters of the RNN. Even if the results shown are interesting, they would be more convincing if the authors would show the performance metrics (RSME and MAPE) in every fold of the cross-validation. Then, select the best performing hyperparameters and apply the RNN to the test set that the model has never seen before to compute out-of-sample performance metrics.
Response: I have supplemented the original text by dividing the data into training set, validation set and test set, and supplemented the prediction effect of different data sets, and replaced the original diagram with a more intuitive Taylor diagram.
- Also, concerns can be raised about the statistical significance of the results in Table 2. It does not currently show that EEMD-Adam-GRU always outperform the other models. In other words, as an example, it is hard to tell if the difference in performance metrics between EEMD-Adam-GRU and AR is large or not. To be more convincing, a suggestion could be to apply the model on multiple time series of hydropower generation (e.g. other countries), then compute the average and standard deviation of the performance metrics. Then, a t-test could show that the performance metrics of EEMD-Adam-GRU is statistically higher than the other models.
Response: The prediction comparison part has been greatly modified. On the one hand, Taylor chart is introduced to compare the prediction performance of different models more intuitively. On the other hand, the original test data range is adjusted and verification set is introduced. Through the comparison between verification set and test set, the validity and generalization of the proposed model are fully illustrated.
- The comparison to other benchmark models in Table 2 looks a little bit unfair. For instance, is EEMD also applied before feeding inputs to the LSTM ? If yes, then LSTM should be called EEMD-LSTM for consistency. If not, it is likely that LSTM might perform worse because it is fed a more noisy feature.
Response: LSTM model has been replaced by EEMD-LSTM in the comparison model, which can reduce the impact of signal noise on the prediction performance of LSTM model, and can be compared with the model proposed in this paper to analyze the effectiveness of different depth learning algorithms.
- Another suggestion would be to reorient the tone of the article towards hydropower generation instead of economic benefits. The contribution of the article about the forecasting of hydropower generation is interesting but it makes an overly simplistic assumption that the price is constant. The discussion about the economic benefits is also interesting but should be presented as an illustration rather than a main result. Hence, the title of the article should be more oriented towards hydropower generation rather than economics.
Response: As there are very few economic calculation data related to the economic benefits of hydropower, the calculation of economic benefits is relatively simple, but it can provide a reference data for managers and researchers. The title has been modified in this paper, with hydropower generation as the main research content in the front of the title, and economic benefits as the secondary research content in the back.
- Limit of the proposed approach: hydropower generation is strongly influenced by the availability of water, hence precipitations. A proper forecasting of the hydropower generation should account for the forecasting of these precipitations.
Response: There are two main ways to forecast hydropower generation. One is the input-output relationship you mentioned. The input set is an influencing factor, including rainfall. However, the influencing factors of hydropower generation include not only rainfall, but also many complex factors such as power system regulation. Therefore, the prediction effect based on the input-output relationship may be unstable. Another way is to regard hydropower generation as a time series in this paper, so that the influence of complex factors is not considered. When the number of data samples is enough, the change rule of data can be deeply mined through the time series prediction model.
- For replication purposes, it would be useful to add more details about the architecture and the training of the RNN. In particular:
a.Architecture : how many layers, neurons are used in the RNN? What are the activation functions used ?
b.What kind of regularization (dropout, early stopping with checkpoints,…) technique are used to avoid overfitting?
c.What is the shape of the training loss, and the test loss? (this is related to point 1.)
d.For EEMD, what is the value of N?
Response: I have added the model parameters that need to be supplemented in the article.
- To assess the scalability of the results presented in the paper, a comparison of CPU times between the different methods would be of interest.
Response: Because the server is used in the model calculation, the CPU configuration is good, and the speed of different models is relatively fast during the calculation, there is no obvious difference. Therefore, the calculation time is not added as the comparison content in this paper.
Minor
- The literature has various ways to solve the mode mixing problem (see e.g. “Causes and classification of EMD mode mixing (2019)”). Why is EEMD chosen instead of another method? If it is better in this context, add some details.
Response: EEMD is an improved empirical mode decomposition method, which can effectively mine different frequency components of signals. The combination with deep learning can improve the accuracy of deep learning prediction. In the introduction, the above statement and literature support are added.
- Line 45-49. In Table 2, the performance metrics of some of the models (e.g. LSTM) are not as bad as what is described in the introduction. Also, how does EEMD-Adam-GRU compare to the prediction accuracy of the models mentioned in the introduction (but not discussed in Table 2)?
Response: The literature in the introduction has been modified, and EEMD-LSTM has been added to the comparison model to ensure the consistency of the methods mentioned in the introduction and the case analysis.
- Equation (7) : Shouldn’t IMF be indexed with i ?
Response: I have revised the express error.
- Line 168 : Small comment : GRU is not always more effective. It depends on the type of application, and the size of the dataset. In a paper from Google researchers (see “An Empirical Exploration of Recurrent Network Architectures” (2015)), they find that GRU outperforms LSTM on all tasks except for language modelling.
Response: I have added the reference” An Empirical Exploration of Recurrent Network Architectures” in the article and add the small comment.
- “Economic benefits” is a vague term. A suggestion would be to refer to “market size” instead of “economic benefits”.
Response: The term "market size" relatively contains more content. This paper mainly focuses on the direct economic benefits brought by hydropower, so the economic benefits are more consistent with the research content of this paper.
- Why does Figure 7 shows predictions up to august 2023 and Figure 6 up to July 2022 (as stated in the text)?
Response: Figure 6 has been deleted and replaced by the Taylor plot. Since the proposed model is a time series prediction model, it is possible to predict future data in advance after the model is built. This paper attempts to predict the data in the next year.
Typos & lack of clarity
1.Line 16 : GRU stands for Gated Recurrent Unit, not Gate Recent Unit.
Response: I have revised the problem.
- Line 33 : Source missing. Also, “hydropower development” requires a definition.
Response: I have added the definition of hydropower development rate in the introduction.
- Line 63 : Citation format has changed.
Response: I have revised the problem.
- Line 86 : Citation missing for “Huang”.
Response: I have revised the problem.
- Line 102 : “instantaneous average m(t)” requires a definition.
Response: I have revised the relevant statement.
- Line 140 : Grammar needs to be checked.
Response: I have revised the relevant statement.
- Line 177 : Citation missing for EEMD.
Response: I added the reference in the corresponding position.
Reviewer 3 Report
This manuscript, water-1986035-peer-review-v1- entitled "Prediction model of economic benefits from hydropower generation based on EEMD-Adam-GRU," is well written and has potential, but it should be more organized. This research investigates the amount of hydropower generation in China and calculates the corresponding economic benefits with high precision; EEMD adaptive moment estimation (Adam) and GRE neural network are integrated.
In my opinion, a careful revision of the English language should be carried out as there currently are some unclear sentences. The study seems to be well-designed. The methodology and results are technically sound. Discussions on the scientific and practical values of the study, the limitations of proposed models, and future work are meaningful. I recommend accepting this manuscript after revision. The main concerns are as follows:
1) The title section should be edited and rewritten since it is too general, and it is better to clarify the model type.
2) Quantitative results should be provided in the abstract to make it more comprehensive. The results of the models Should be added in the abstract section. Also, The main aim of the study should be clearly mentioned in the abstract.
3) Some abbreviations in the paper have already not been addressed in the text, like EMD-LSTM.
4) More literature review about the other methods is needed. The manuscript could be substantially improved by relying and citing more on recent literature about contemporary real-life case studies of sustainability and/or uncertainty, such as the followings.
· Samani, S., Vadiati, M., Azizi, F., Zamani, E., & Kisi, O. (2022). Groundwater Level Simulation Using Soft Computing Methods with Emphasis on Major Meteorological Components. Water Resources Management, 36(10), 3627-3647.
· Vadiati, M., Rajabi Yami, Z., Eskandari, E., Nakhaei, M., & Kisi, O. (2022). Application of artificial intelligence models for prediction of groundwater level fluctuations: Case study (Tehran-Karaj alluvial aquifer). Environmental Monitoring and Assessment, 194(9), 1-21.
5) Providing a comprehensive flowchart is highly recommended by researchers, so please add a flowchart representing the methodology in the paper.
6) What are other feasible alternatives for the case study? What are the advantages of adopting this case study over others? How will this affect the results? The authors should provide more details on this.
7) Please provide all software used in this study with the library or packages.
8) It is important to give a better description of the samples and the sampling protocol since we are trying to understand the data variability. What are the advantages of adopting these parameters over others in this case? How will this affect the results? More details should be furnished.
9) It is better to add more error criteria to understand the model's ability better.
10) The authors should explain more about Fig. 7 in the discussion section.
11) It seems that conclusions are observations only, and the manuscript needs thorough checking for explanations given for results. The authors should interpret the results argument.
Author Response
Dear editors and reviewers:
Thank you for your letter and the reviewers’comments on our manuscript entitled " Prediction model of economic benefits from hydropower generation based on EEMD-Adam-GRU". Those comments are very helpful for revising and improving our paper, as well as the important guiding significance to other research. We have studied the comments carefully and made corrections which we hope meet with approval. The main corrections are in the manuscript and the responds to the reviewers’comments are as follows.
Replies to the reviewers’comments:
Reviewer #3:
- The title section should be edited and rewritten since it is too general, and it is better to clarify the model type.
Response: I modified the title and added "fusion model" to describe the type of prediction model.
- Quantitative results should be provided in the abstract to make it more comprehensive. The results of the models Should be added in the abstract section. Also, The main aim of the study should be clearly mentioned in the abstract.
Response: I have modified the abstract and added the quantitative results and the main aim of the study.
- Some abbreviations in the paper have already not been addressed in the text, like EMD-LSTM.
Response: I have supplemented the full name of the first abbreviated model name in the article.
- More literature review about the other methods is needed. The manuscript could be substantially improved by relying and citing more on recent literature about contemporary real-life case studies of sustainability and/or uncertainty, such as the followings.
Response: I have added more literature review about the other methods in the article.
- Providing a comprehensive flowchart is highly recommended by researchers, so please add a flowchart representing the methodology in the paper.
Response: The EEMD-Adam-GRU fusion model construction flow chart is added in Fig.3.
- What are other feasible alternatives for the case study? What are the advantages of adopting this case study over others? How will this affect the results? The authors should provide more details on this.
Response: In the case study, the hydropower generation is selected as the prediction object, and the proposed model can be applied to the rest of the time series of water conservancy projects in the subsequent research to expand the application scope of the model. As an important function of hydraulic engineering, power generation has made a great contribution to the growth of the national economy worldwide. Therefore, it is of practical engineering significance to analyze and predict the hydropower generation and its economic benefits. This fusion model makes full use of GRU neural network's advantages in predicting periodic time series. although there are many kinds of research objects that can be selected, as long as it is a time series, the model proposed in this paper can be used for prediction, but the prediction effect may be different due to different signals.
- Please provide all software used in this study with the library or packages.
Response: If publish needed, relevant programs and codes will be uploaded as attachments.
- It is important to give a better description of the samples and the sampling protocol since we are trying to understand the data variability. What are the advantages of adopting these parameters over others in this case? How will this affect the results? More details should be furnished.
Response: The selection of model parameters has a great impact on the accuracy of the time series prediction model. This paper uses Adam optimization algorithm to obtain the optimal parameters of the model, which avoids the impact of random selection of model parameters on the prediction effect. In order to better demonstrate the effectiveness of the model comparison in this paper, Table 2 is added, which lists various selection parameters of all comparison models.
- It is better to add more error criteria to understand the model's ability better.
Response: In order to better compare the prediction errors of different models, Taylor plot is an effective method to intuitively display the model error. The original diagram is intended to show the change trend of prediction performance of different models. Therefore, the graph is replaced by the Taylor plot.
- The authors should explain more about Fig. 7 in the discussion section.
Response: I add more statement about Fig.7 in the discussion section.
- It seems that conclusions are observations only, and the manuscript needs thorough checking for explanations given for results. The authors should interpret the results argument.
Response: I have checked the conclusion and revised some statement.
Round 2
Reviewer 1 Report
Accept in present form
Reviewer 3 Report
The authors addressed all comments and did a great effort to improve the paper.